# Heterologous expression, purification, and biochemical characterization of protease 3075 from *Cohnella* sp. A01

**Fatemeh Hashemi Shahraki◉, Narges Evazzadeh◉, Saeed Aminzadeh ◉ ***

Bioprocess Engineering Group, Institute of Industrial and Environmental Biotechnology, National Institute of Genetic Engineering and Biotechnology (NIGEB), Tehran, Iran

◉ These authors contributed equally to this work.
* aminzade@nigeb.ac.ir

## Abstract

Proteases as one of the most significant categories of commercial enzymes, serve nowadays as the key ingredients in detergent formulations. Therefore, identifying detergent-compatible proteases with better properties is a continuous exercise. Accordingly, we were interested in the recombinant production and characterization of protease 3075 as a novel enzyme from thermophilic indigenous *Cohnella* sp. A01. The biochemical and structural features of the protease were probed by employing bioinformatic methods and in vitro studies. The bioinformatics analysis discovered that protease 3075 belongs to the $C_{56}$—*PfpI* superfamily. The protease 3075 gene was cloned and heterologous expressed in *Escherichia coli (E. coli)* BL21. It was found that the enzyme contains 175 amino acids and 525 bp with a molecular weight of 19 kDa. Protease 3075 revealed acceptable activity in the range of 40–80°C and pH 5.5–8. The optimum activity of the enzyme was observed at 70°C and pH 6. The activity of protease 3075 increased about 4-fold in the presence of Tween 80 and acetone, while its activity attenuated in the presence of iodoacetic acid and iodoacetamide. Docking analyses revealed the dominant interaction between Tween 80 and protease 3075, mediated by hydrogen bonds and Van der Waals forces. Furthermore, molecular dynamics simulations (MDS) showed that Tween 80 increased the stability of the protease 3075 structure. Altogether, our data provided a novel enzyme by genetic manipulation process that could have significant industrial applications.

**Data Availability Statement:** All relevant data are within the manuscript and its Supporting Information files.

## 1. Introduction

Today, the biotechnology industry has undergone significant transformation due to the efficiency of enzymes as catalysts, their cost-effectiveness, and biodegradability [1]. Proteases are one of the most important categories of industrial enzymes that account for up to 60% of the global enzyme trade [2–4]. These enzymes are widely used in various industries such as pharmaceuticals, food, leather, detergents, and wastewater treatment [5–7]. Therefore, there is an increasing need for efficient methods to provide such enzymes in high quantities and quality.

**Funding:** The author(s) received no specific funding for this work.

**Competing interests:** The authors have declared that no competing interests exist.

Microorganisms are eligible for producing industrial enzymes due to their desirable performance. However, the preparation of proteases is very challenging due to some limitations in the production pathway [8]. Hence, the introduction of novel and more efficient proteases to improve expression, stability, and resistance to various environmental conditions such as temperature, pH, inhibitory compounds, etc. to optimize vectors, and host strains and ultimately optimize enzyme performance is very important [9]. The catalytic activity and stability of protease at alkaline pH and high temperatures are crucial for industrial bioprocesses. Proteases have a wide spectrum [10]. Despite the study and identification of a wide range of them, reports on thermophilic and even hyperthermophilic proteases are limited. Therefore, genetic engineering to improve the catalytic properties of proteases and the search for new sources of protease enzymes are essential to meet the increasing industrial demands [11]. On the other hand, since the global market is moving towards increased use of proteases with specialized applications, the highest production costs of this category of enzymes are related to downstream processes, separation methods, and purification. Therefore, there is a pressing need to improve production methods and downstream processes to make proteases more cost-effective in industries [12]. This is the main problem of using proteolytic enzymes in industries, which requires further studies to find more suitable conditions and improve the industrial market of proteases [13]. Microorganisms produce protease enzymes with a wider biochemical diversity [14, 15]. One of the major challenges in the commercial use of some enzymes is their limited thermal stability in many industries, including dairy and detergent industries [16]. Although many heat-resistant proteases have been produced by genetic engineering and thermophilic bacteria, there is still a pressing need to identify new proteases that meet industrial requirements such as pH tolerance, temperature range, and high salt concentration. Despite the identification of a wide range of proteases, new, efficient, scalable, and cost-effective strategies are still needed to improve downstream processes and increase the usability of proteases in industrial applications [17, 18]. Moreover, the thermal stability of the protease enzymes is one of the significant specifications since many industrial processes are carried out at high temperatures where many enzymes are not stable [19]. Therefore, in the current study, we tried to introduce a novel thermostable protease with various industrial applications. To end this, recombinant expression and purification of the protease 3075 gene from the thermophilic *Cohnella* sp. A01 was carried out. The recombinant protease 3075 was cloned in the pET26 b (+) vector. *Escherichia coli* (*E. coli*) BL21 was selected as a system for expressing the biologically active form of the recombinant protease. After expression, the expressed protein was purified by Ni-NTA affinity chromatography. To characterize protease 3075 structural and biochemical features, the kinetic parameters, pH profiling, thermal stability, thermodynamics parameters, molecular docking, and molecular dynamics simulation (MDS) were applied to evaluate the potential industrial application.

## 2. Materials and methods

### 2.1. Strains, plasmids, reagents, and materials

*E. coli* strains BL21 (DH5$_\alpha$) was bought from Invitrogen Company (USA). pET26b (+) vector, T4 DNA ligase, Nde I cleavage enzyme, RNase A, Proteinase K, Taq polymerase, and *Pfu* polymerase all were purchased from Fermentas. High-purity plasmid extraction and PCR product kits were prepared from Bioneer. DNA extraction kit was obtained from Peqlab company. All other materials were received by Sigma Aldrich Co.

## 2.2. Methods

**2.2.1. *Cohnella* sp. A01 cell culture and DNA extraction.** Bacterial stock *Cohnella* sp. A01 was cultured in an LB broth medium at pH 6.8. The culture medium was kept for 72 hours in a shaker incubator with a temperature of 58°C and a speed of 180 rpm. 2 ml of culture medium was added to the 50 ml of fresh culture medium. Then was incubated for 5 days with the same conditions. After the turbidity (OD) of the culture medium reached 0.4 at 600 nm wavelength, 5 ml of the medium was centrifuged at 4000 g for 10 minutes. The supernatant solution was discarded and genomic DNA was extracted using the instructions of the DNA purification kit.

**2.2.2. Cloning and heterologous expression of protease 3075 gene.** To isolate the protease gene from the bacterial genome and amplify it, forward (5'GGAATTCCATATGAAGAA AGTCGCTTTCCTG3') and reverse (5'CCGCTCGAGAGACTGAGGGACGGG3') primers without stop codon were designed using the nucleotide sequence of the beginning and end of the gene. The restriction enzyme cleavage site (*Nde* I and *Xho*I) was inserted into the sequence of the primers for transferring the gene to the expression vector pET-26 b (+). To better the performance of the restriction enzymes, before the cleavage site, a few nucleotides were added to the sequence of primers as the site of enzyme placement. After designing the primers www.expasy.org was used to ensure the correctness of the gene reading format during translation, and the gene was translated from the start code to the end code [20]. The direct primers contained 24 nucleotides and a cleavage site for the NdeI enzyme. The reverse primer was 21 nucleotides with a cleavage site for the XhoI enzyme. The primers were made by Tekapozist. The PCR product (50 ng) was cloned into pET-26b(+) plasmid (25 ng/μl) by $T_4$ DNA ligase (10x) and kept for three hours at 22°C. The Recombinant plasmid (25 ng/ μl) was transferred to *E. coli* DH5α bacteria (OD ~ 0.5) by heat shock method. A colony of bacteria was cultured on an LB broth plate with kanamycin and kept at 37°C for 16 hours. To induce gene expression, IPTG (1 mM) was added to the culture medium and kept at 27°C and 50 rpm for 16 hours. The expression level (1, 3, 6, and 16 hours after induction with IPTG) was checked by SDS-PAGE. After overnight cultivation, centrifugations at 8000 in 20 minutes were performed. Then, the pellet was dissolved in the binding buffer and sonicated with 0.5 pulses per minute, during 4-time cycles of 45 seconds and a minute of rest. The obtained supernatant was centrifuged for 40 minutes at 8000 rpm at 4°C. Finally, the supernatant containing protein and precipitate after sonication was subjected to SDS-PAGE.

**2.2.3. Recombinant enzyme purification.** *2.2.3.1. Ni-NTA affinity chromatography*. For the recombinant protein purification, Ni-NTA resin affinity chromatography with a volume of 1 ml Nickel Sepharose Resin was prepared. First, the protein fraction (4 ml) was dissolved in the binding buffer. To equilibrate the column, the resin was washed with an equilibration buffer (0.05% Tween 20, 50 mM $NaH_2PO_4$, 10 mM Imidazole, and 300 mM $NaCl^+$ at pH 7). Then, the chromatography column resin was cleansed with a buffer 3 times (0.05% Tween 20, 50 mM $NaH_2PO_4$, 300 mM $NaCl^+$, at pH7) and elution buffer 4 times (0.05% Tween 20, 300 mM $NaCl^+$, 30 mM Imidazole, and 50 mM $NaH_2PO_4$ at pH 7). The protein solution was loaded onto the column 6 times. To remove other proteins, an elution buffer (0.05% Tween 20, 50 mM $NaH_2PO_4$, 250 mM Imidazole, and 300 mM $NaCl^+$ at pH 7) was added to the column. After washing the other proteins from the column, by increasing the concentration of imidazole, the recombinant protein was extracted from the column in pure form. The recombinant protein was eluted from the column using an elution buffer. The purified proteins were dialyzed over 16 hours at 4°C in potassium phosphate buffer (50 mM). Samples were prepared from all stages and SDS PAGE was put on them. Determination of concentration was done by the Bradford method [21].

**2.2.4. Western blotting.** To confirm the presence of the recombinant protein on SDS-PAGE, western blotting was performed by HRP-POLYHIS conjugated antibody against the terminal histidine of the protein. To end this, the protein fractions were transferred to a PVDF membrane. The mentioned membrane was placed at 25°C over 5 h with TBST (0.05% Tween 20, 0.14 mM NaCl$^+$, 25 mM Tris–HCl, and pH 7.4) including 5% BSA. The transferred membrane was washed by TBST 3 times. Then, was incubated for 3 hours at 37°C in a 1:2000 diluted monoclonal anti-poly-His peroxidase. After 3 washes using TBST, the target protein appeared by prospering a spot with H$_2$O$_2$ and 4-chloro-1-naphthol.

**2.2.5. Enzyme activity assay.** To study the protease 3075 activity and substrate specification, the enzyme activity was evaluated by a variety of substrates, including casein, azo-casein, azo-albumin, and gelatin. The substrates were dissolved in potassium phosphate buffer (50 mm). The reaction mixtures were prepared by mixing the protease 3075 (120 μl) with 480 μl of azo-casein. Then, 120 μl of the enzyme with 1ml azo-albumin was incubated at the optimum temperature for 30 minutes. The next station started with incubation of 70 μl casein with 50 μl of the protein, and 70 μl gelatin with 50 μl of the protein respectively, over 20 minutes at 70°C. All the reaction was stopped by adding a fresh and cold trichloroacetic acid (TCA, 10%). Finally, the samples were centrifuged at 12000 g and the supernatant absorbance was monitored. Kinetic parameters (K$_m$, $V_{max}$, $K_{cat}$, and $K_{cat}/K_m$) were characterized according to the Michaelis-Menten model [22]. The protease 3075 activity was studied in the presence of various concentrations of casein (5–60 mg.ml$^{-1}$) as substrate at the optimum temperature. The enzyme activity assay was carried out with 0.9 mg.ml$^{-1}$ of the purified protein in 3 times repeats.

**2.2.6. Zymography.** The casein Zymography analysis was done on SDS-PAGE as described by Garcia-Carreno et al. [23]. In this method, enzyme activity was evaluated in non-denaturing polyacrylamide gel. The non-denaturing polyacrylamide gel containing the substrate was prepared, then 20 μl of the enzyme was poured into a well and the gel was connected to an electric current generator with a potential difference of 80 V. After 16 hours of electrophoresis, the gel was treated for 2 hours in potassium phosphate buffer at optimal pH and calcium ion (2 mM). Then, Coomassie blue dye was added to the gel for 15 minutes.

**2.2.7. Temperature and pH effects on the stability and enzyme activity.** To investigate the effect of temperature on the stability and activity of protease 3075 in the presence of 1% casein, the purified and dialyzed enzyme was incubated for 3 hours in the temperature range of 10 to 80°C. To evaluate the effect of pH on enzyme activity, a pH profile was prepared in the range of 3 to 10. A mixture of pure enzyme and buffer with a final pH of 4 to 10 was prepared. The reaction mixture was incubated for 3 hours at 4°C. Then, the enzyme activity was measured at the optimal temperature and pH.

**2.2.8. Effects of ions, organic solvents, surfactants, and chemical materials on the proteolytic activity.** The impact of metal ions on protease 3075 activity was investigated in the presence of 2 and 5 mM concentrations of MnCl$_2$, MgCl$_2$, CaCl$_2$, ZnCl$_2$, NaN$_3$, NaCl$_2$, and CuCl$_2$. The effect of surfactants as organic compounds was monitored using the 1% and 2% (V/V) concentrations of Tween 80, Tween 20, triton X$_{100}$, H$_2$O$_2$, and Sodium Dodecyl Sulfate (SDS) in the reaction mixture of enzymes. The effect of organic solvents on the protease activity was evaluated using 1% and 2% (v/v) concentrations of four different solvents such as methanol, ethanol, isopropanol, and acetone. To assess the impact of chemical compounds, the enzyme activity was evaluated in the presence of 2 and 5 mM concentrations of iodoacetic acid (IAA), iodoacetamide (IAM), phenylmethylsulphonyl fluoride (PMSF), and ethylenediaminetetraacetic acid (EDTA). The activity of protease 3075 was considered 100%, and the residual enzyme activity in the presence of mentioned compounds and solutions was measured.

**2.2.9. Thermodynamic parameters study.** The irreversible thermal inactivation and enzyme activation energy can be considered as the elementary relations associating the alterations in the enthalpy ($\Delta H$), Gibbs free energy ($\Delta G$), and entropy ($\Delta S$). The induced alterations in the entropy and enthalpy of activation energy ($\Delta H^{\ddagger}$, $\Delta S^{\ddagger}$) to the shape of enzyme-substrate complex [ES] were computed by using the Arrhenius Eqs (1–4) [24]:

$$K_{cat} = K_B T/hK*_{cat} \tag{1}$$

$$\Delta G^{\ddagger} = -RT \ln K*_{cat} \tag{2}$$

$$\Delta H^{\ddagger} = E_a - RT \tag{3}$$

$$\Delta G^{\ddagger} = \Delta H^{\ddagger} - T\Delta S^{\ddagger} \tag{4}$$

$k_B$ = Boltzmann constant (R/N, is $1.38 \times 10{-}23$ J.K$^{-1}$)
T = Temperature in Kelvin
h = Planck's constant ($6.62607015 \times 10^{-34}$ m$^2$ kg. S$^{-1}$)
$\Delta G^{\ddagger}$ = Gibbs free energy of activation energy
R = Gas constant (8.314 J. K mol$^{-1}$)
$\Delta H^{\ddagger}$ = Change in the enthalpy
Ea$^{\ddagger}$ = Energy of activation
$\Delta S^{\ddagger}$ = Change in the entropy

**2.2.10. Structural and bioinformatics analysis.** The bacterial genome of *Cohnella* sp. A01 was isolated from a shrimp breeding pond in Abadan, Iran. Then the sequence and possible genes were predicted by Baseclear Netherlands company [25]. The biochemical properties of protease 3075 were analyzed using the online Expasy translate server (http://web.expasy.org/translate). Biochemical features including instability index, isoelectric point, molecular weight, aliphatic coefficient, and protein net charge were monitored by the Expasy ProtParam tool (http://web.expasy.org/protparam) [26]. The presence of disulfide bonds was determined using the http://disulfind.dsi.unifi.it website. Secondary and tertiary structures were predicted by http://www.sbg.bio.ic.ac.uk/pHyre and http://swissmodel.expasy.org, respectively. The final structure model of protease 3075 was prepared by Easymodeller V.2 software. Furthermore, To explore homology across different species, the amino acid sequence of the novel *Cohnella* sp. A01 protease 3075 (accession number AXE74952.1) was inputted into the GenBank alignment tool of Basic Local Alignment Search Tool (BLAST) available at https://blast.ncbi.nlm.nih.gov/Blast.cgi and was performed a protein BLAST. Subsequently, all subject sequences were downloaded and imported into MEGA11 software for multiple alignment. Phylogenetic trees were then constructed using the neighbor-joining (NJ) method within MEGA11 for both AXE74952.1- *Cohnella* sp. A01 and the retrieved sequences.

**2.2.11. Identification of crystallographic models.** The amino acid sequence of the *Cohnella* sp. A01's protease was evaluated by www.ncbi.nlm.nih.gov website. P-BLAST was performed to find similar sequences and apply e-Value to PDB of possible similar amino acid sequences. The Ramachandran plot was drawn by employing PROCHECK (http://services.mbi.ucla.edu/PROCHECK) at the SAVE server (http://services.mbi.ucla.edu/SAVES) to monitor the validation of the prepared 3D structure. The website https://prosa.services.came.sbg.ac.at/prosa.pHp was applied to determine the Z-Score and protein energy balance [27].

**2.2.12. Molecular docking.** The patterned protease 3075 was docked by Tween 80, as a type of surfactant. The molecular docking technique was employed to examine the dynamic behavior of the created patterns. To end this the Auto Dock 4.2.6 software was employed

[28, 29]. The 3D structure of Tween 80, as a ligand, was achieved from the PubChem site. After optimizing the achieved structures by Chimera V 1.13.1 software, the crystal structures were saved in a PDB format and determined the best-docked model [30, 31]. Hyper-Chem 8.0.6 software was employed for energy minimization. To identify the interaction style, a docking survey was done in two different steps; I) initial docking and II) final docking. Then, the best docking mood was picked out. Important factors including orientations and random torsions were considered for the protease [32]. In the molecular docking analysis, the grid box size was set to 126 Å × 126 Å × 126 Å. The distance between each atom in the protein and the cubic box wall was set at 1 Å. For the Van der Waals interactions 0.1 nm cutoff was considered. Docking studies were performed by adding 7808 water molecules during 100 ns. Over the docking process, the water model was $TIP_{4P}$.

**2.2.13. Molecular dynamic simulation.** The Gromacs V 4.6.5 program was used to study the molecular dynamics simulations of the protease 3075-Tween 80 complex. Amber 99 force field was applied during the simulation process. The free enzyme and enzyme-ligand complexes were located in the simulated cubic box. The amount of system energy was minimized by using the Brendes algorithm to dispose of the improper interactions. A cubic box with dimensions of 3.197 Å × 3.197 Å × 3.197 Å was applied to analyze the simulated system. Then the complex was put in the center of the cubic. Similar to the docking procedure, over the simulation studies, the Brendes algorithm was applied to energy minimization. The dynamic simulations were carried out over 100 ns.

## 3. Results

### 3.1. Protease 3075 expression in bacteria

Protease 3075 was expressed heterologously in *E. coli* BL21. Plasmid pET-26 b (+) was employed for the transformation of *E. coli* BL21. To end this, the coding gene of protease 3075 was driven from the genomic DNA of *Cohnella* sp. A01. Then, was amplified by PCR and inserted into the expression plasmid pET-26 b (+). The authenticity of the recombinant plasmid was monitored by applying PCR with specific primers, sequencing, and double digestion (S1 Fig). The SDS-PAGE results demonstrated intracellular collection of protein with a molecular weight of about 19 kDa. The mentioned protein was in a soluble form (Fig 1A).

### 3.2. Purification, zymography, and western blotting of protease 3075

His tag was added to the C-terminal of the recombinant protease 3075. Then, the recombinant enzyme was purified by the Ni-NTA column. The final yield of the column was 63%. The SDS-PAGE result of the purification process is shown in Fig 1A. Zymography was done to investigate the protease 3075 activity by Casein substrate (Fig 1B). Furthermore, the enzyme activity and efficiency of purification are listed in Table 1. The zymography results revealed the protease 3075 activity at the molecular mass of 19 kDa (Fig 1C). Western blot results by HRP-POLYHIS conjugated antibody showed a similar molecular mass of the enzyme.

### 3.3. Enzyme activity assay and kinetic parameters

The enzyme activity assessment was carried out with a 0.9 mg.ml$^{-1}$ concentration of protease 3075. The substrate specification of protease 3075 was investigated by comparing well-known protease substrates. Casein, azo-casein, azo-albumin, and gelatin were selected as protease 3075 substrates. The protease activity was determined by measuring absorbance at 280 nm. The absorbance intensity values revealed the maximum absorption in the presence of casein and gelatin. While azo-albumin and azo-casein showed the minimum absorbance. Moreover,

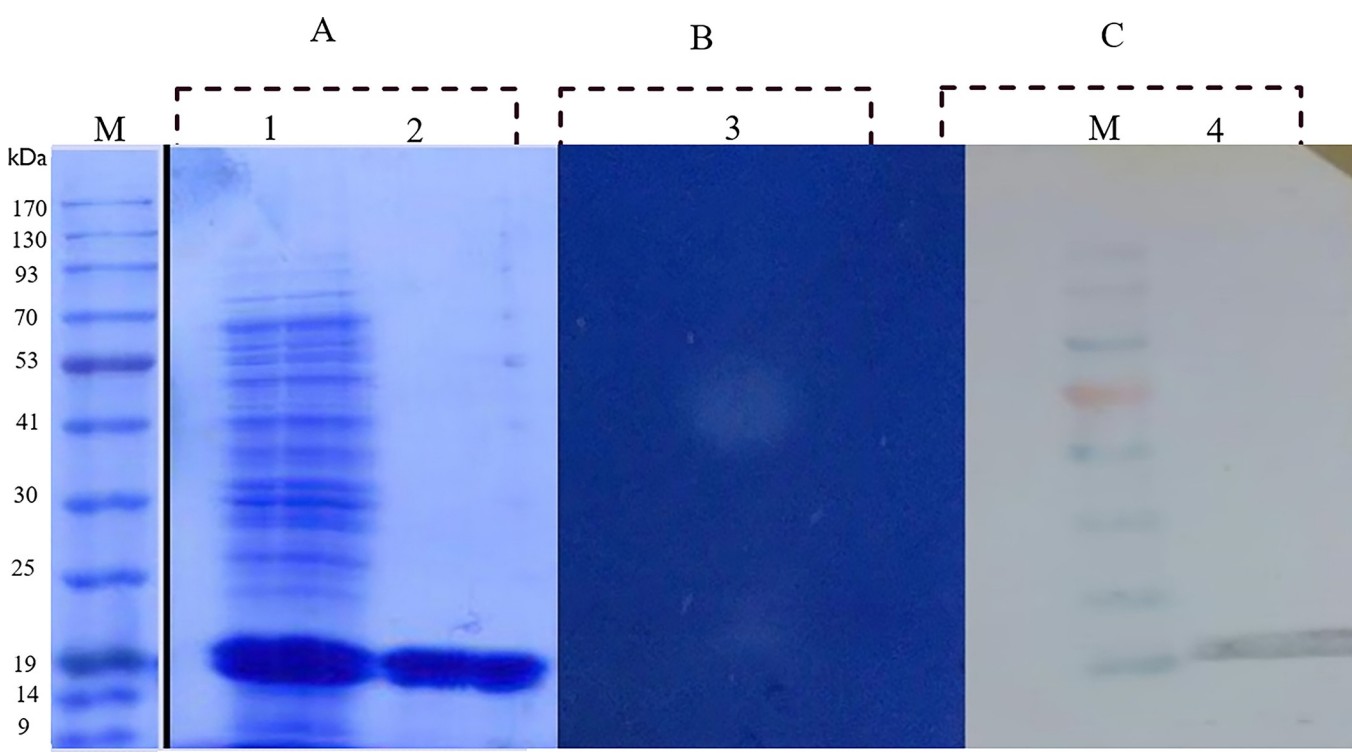

**Fig 1. SDS-PAGE, zymogram, and western blot analysis of recombinant protease 3075.** (A): SDS-PAGE analysis. Line 1: crude extract after sonication. Line 2: crude extract purified by Ni-NTA. (B, line 3): Native-PAGE zymography analysis of the crude extract of purified protease 3075 by Ni-NTA affinity column and creating the digested area. (C, line 4) Western blot analysis of purified protease 3075 by Ni-NTA chromatography. M refers to marker.

the kinetic parameters of protease 3075 were calculated in the presence of casein substrate by the Michaelis-Menten plot (Fig 2). The values of $K_m$ and $V_{max}$ were determined about 58.21 mg.ml$^{-1}$ and 0.041 mM, respectively (Table 2).

## 3.4. Measurement of temperature and pH stability and the effects of them

The protease activity was investigated at different temperatures and pH. The activity of protease 3075 was studied at 10–90˚C. Based on Fig 3A, at the temperature range of 40–80˚C, the protease showed more than 50% activity. Moreover, the optimal activity of the enzyme was obtained at 70 C. Examining enzyme activity at different pH (3–10) showed that the optimal pH of protease 3075 was 6. More than 40% activity at pH 5–8 was seen (Fig 3B). The enzyme was kept at temperatures of 10–90˚C (Fig 3C), and pH 4–9 (Fig 3D) over 90 minutes. Then the protease activity was investigated under optimal conditions (70˚C, pH 6). According to Fig 3C, the enzyme maintained more than 70% of its activity at 10–70˚C.

The thermal stability of protease 3075 was studied after incubation at 60, 70, and 80˚C over 1, 2, and 3 hours at optimal conditions. The obtained information showed that more than 50% of the enzyme activity was preserved after 3 hours at 60 and 70˚C (Fig 3E). Furthermore,

**Table 1. Protein-specific activity, total activity, and the amounts of the yield of the purified enzyme by Ni-NTA chromatography.**

| Step | Volume (ml) | protein (mg.ml$^{-1}$) | Total Protein (mg.ml$^{-1}$) | Total Activity (U.ml$^{-1}$) | Specific Activity (U.mg$^{-1}$ protein) | Yield (%) |
|---|---|---|---|---|---|---|
| Crude Extract | 4 | 2.8 | 11.2 | 3.92 | 0.35 | 100 |
| Ni-NTA | 4 | 0.9 | 3.6 | 1.44 | 0.4 | 63 |

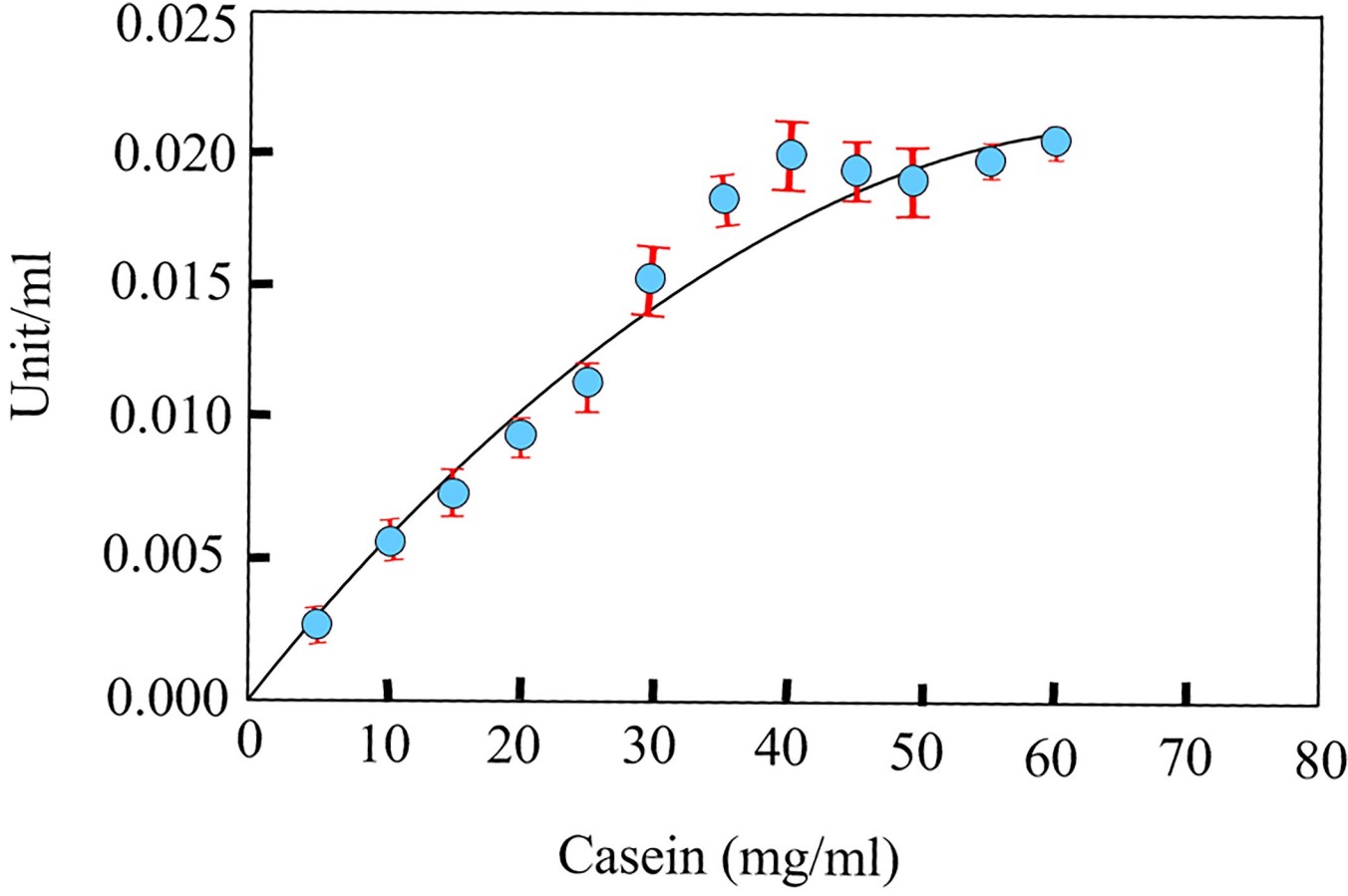

**Fig 2. Michaelis-Menten diagram of protease 3075 activity in the presence of different concentrations of casein.**

Enzyme stability studies at pH 5 and 9 at different times showed that 50% of protease 3075 activity was retained after 2 hours in this pH range (Fig 3F).

### 3.5. Determination of the protease 3075 thermodynamic parameters

The activation energy ($E_a^{\ddagger}$) of protease 3075 was calculated at 0.023 kJ.mol$^{-1}$ using the Arrhenius equation (Fig 3G). Gibbs free energy ($\Delta G^{\ddagger}$), enthalpy ($\Delta H^{\ddagger}$), and entropy ($\Delta S^{\ddagger}$) of the reaction at optimum temperature were 67.30 kJ.mol$^{-1}$ and -2.82 kJ.mol$^{-1}$, and -0.20 kJ.mol$^{-1}$.k$^{-1}$, respectively (Table 2). All thermodynamic parameters were calculated at optimum temperature. The low obtained values of enthalpy, and entropy parameters at 70°C revealed that the reaction occurs spontaneously.

**Table 2. The kinetic and thermodynamic parameters of protease 3075.**

| parameters | Volume |
|---|---|
| $V_{\mathrm{max}}$ (mg.ml-1) | 58.21 |
| $K_{\mathrm{m}}$ (mM) | 0.041 |
| $E_a^{\ddagger}$ (kJ.mol$^{-1}$) | 0.023 |
| $\Delta G^{\ddagger}$ (kJ.mol$^{-1}$) | 67.30 |
| $\Delta H^{\ddagger}$ (kJ.mol$^{-1}$) | -2.82 |
| $\Delta S^{\ddagger}$ (kJ.mol$^{-1}$.k$^{-1}$) | -0.20 |

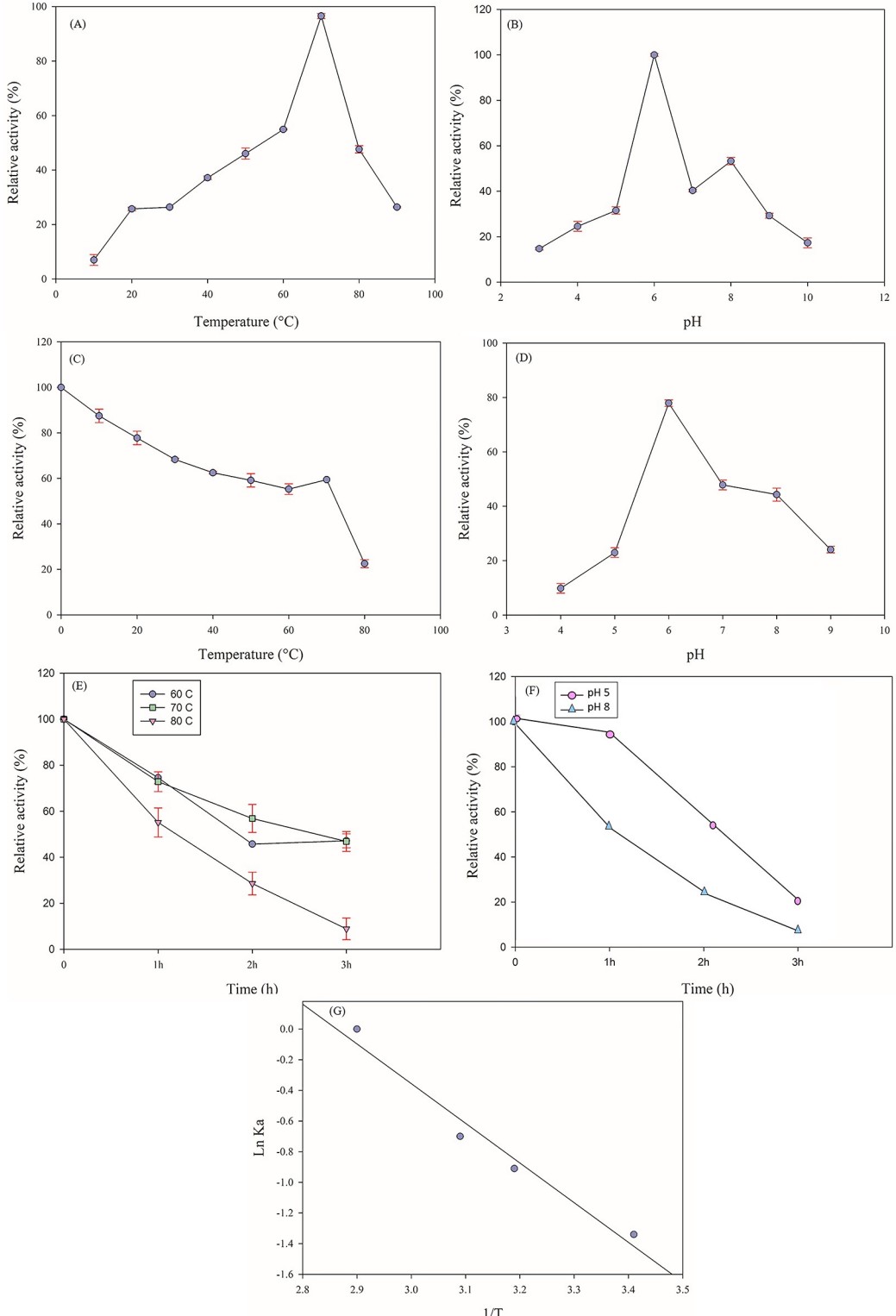

**Fig 3. The impacts of different temperatures and pH on the recombinant protease 3075.** (A): Effect of temperature on the enzyme activity. (B): pH profile and analysis of protease 3075 activity. (C): Effect of temperature on the protease stability. (D): Effect of pH on the enzyme stability. (E): Thermal stability of protease 3075 at different times. (F): Enzyme pH stability assay (G): Arrhenius plots of activation energy (Ea‡).

### 3.6. Effects of surfactants, metal ions, chemical compounds, and organic solvent on the proteolytic activity of protease 3075

Surfactants as organic compounds decrease the surface tension. In this study, the effects of tween 80, tween 20, triton x100, $H_2O_2$, and Sodium Dodecyl Sulfate (SDS) were investigated on the protease 3075 activity under optimum pH and temperature. According to the obtained data, tween 20, triton x100, and Sodium Dodecyl Sulfate inhibited the enzyme activity, while tween 80 increased the enzyme activity 4 fold (Fig 4A). Metal ions including $MnCl_2$, $MgCl_2$, $CaCl_2$, and $ZnCl_2$, $NaN_3$, $NaCl_2$, and $CuCl_2$ in the selected concentrations showed different effects on the protease activity. $MnCl_2$ and $NaCl_2$ caused reduced enzyme activity. In contrast, $CaCl_2$ at 2 mM concentration increased enzyme activity 2-fold (Fig 4B).

The effect of 2 and 5 mM concentrations of chemical compounds including iodoacetic acid (IAA), iodoacetamide (IAM), phenylmethylsulphonyl fluoride (PMSF), and ethylenediamine-tetraacetic acid (EDTA) on the enzyme activity was investigated at pH 6 and 70°C. It was determined that iodoacetic acid and iodoacetamide were cysteine protease inhibitors. While PMSF and EDTA reduced enzyme activity by about 50% (Fig 4C). Moreover, the effect of organic solvents such as methanol, ethanol, isopropanol, and acetone in the concentrations of

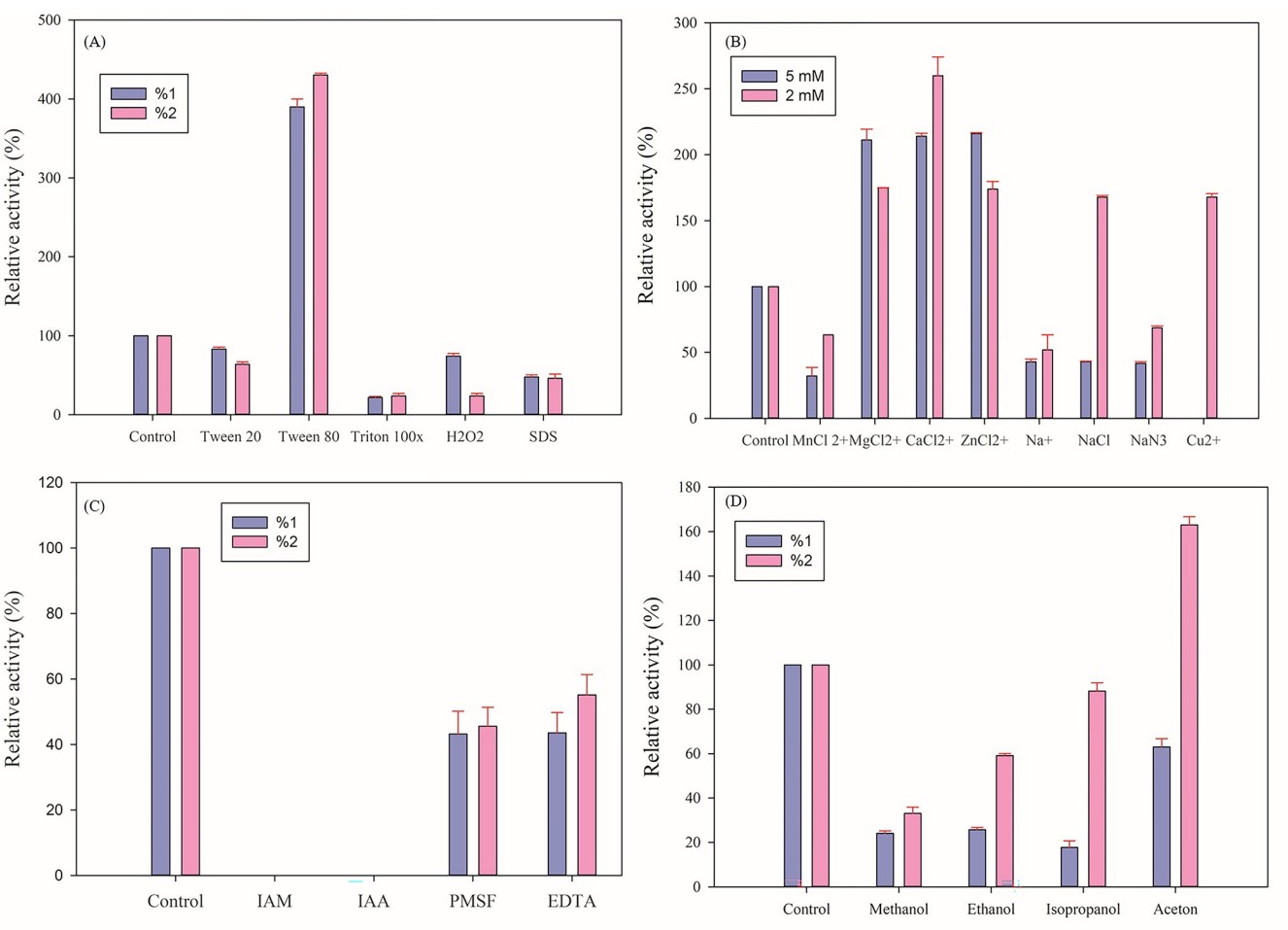

**Fig 4. Effect of different compounds on the proteolytic activity of protease 3075.** (A): effects of various surfactants, (B) metal ions, (C): chemical compounds, and (D): organic solvents on the proteolytic activity of protease 3075 at 1 and 2% concentrations under optimal pH and temperature. All tests were repeated three independent times with a standard deviation.

**Table 3. Effects of surfactants, metal ions, chemical compounds, and organic solvent on the proteolytic activity of protease 3075.**

| Ingredients | Relative activity | |
|---|---|---|
| **Surfactants** | **1%** | **2%** |
| Control | 100±0.00 | 100±0.00 |
| Tween 80 | 390%±10 | 430.33%±2.51 |
| Tween 20 | 63.66%±3.21 | 83%±2.64 |
| Triton $X_{100}$ | 2.66%±1.52 | 23.66%±3.21 |
| SDS | 46%±5.29 | 47.66%±2.51 |
| $H_2O_2$ | 23.66%±3.21 | 74%±3.60 |
| **Metal ions** | **2mM** | **5mM** |
| Control | 100±0.00 | 100±0.00 |
| $MnCl_2$ | 32.01%±3.64 | 63.25±0.00 |
| $MgCl_2$ | 174.89±0.143 | 211.31%±6.64 |
| $CaCl_2$ | 214.10±2.28 | 260.02±14.15 |
| $ZnCl_2$ | 173.98±5.81 | 216.07±0.76 |
| $NaN_3$ | 41.70±1.35 | 68.61±1.57 |
| NaCl | 43.80±1.32 | 167.84±1.35 |
| **Chemical compounds** | **2mM** | **5mM** |
| Control | 100±0.00 | 100±0.00 |
| IAA | 00.00±0.00 | 00.00±0.00 |
| IAM | 00.00±0.00 | 00.00±0.00 |
| PMSF | 43.21±6.96 | 45.56±5.75 |
| EDTA | 43.57±2.02 | 55.16±6.21 |
| **Organic solvent** | **1%** | **2%** |
| Control | 100±0.00 | 100±0.00 |
| Methanol | 24.03±1.14 | 33.05±2.79 |
| Ethanol | 25.70±1.06 | 59.10±0.90 |
| Isopropanol | 17.72±2.97 | 88.20±3.72 |
| Acetone | 63.03±3.67 | 163.03±3.67 |

1 and 2% on protease 3075 activity was measured under optimal conditions. It was found that 2% acetone increased enzyme activity about 150-fold (Fig 4D). The effects of all reagents are summarized in Table 3.

## 3.7. Bioinformatic analysis

### 3.7.1. Biochemical, structural, and phylogenetic tree analysis of protease 3075.
The DNA sequence was translated to a protein with 175 amino acids. Biochemical properties of protease 3075 are reported in Table 4. The protease secondary structure predicted by www.bioinf.cs.ucl.ac.uk/psipred. It was found that the peptide sequence of protease 3075 contained 9 β-Sheets and 5 α-Helix (S2 Fig). The tertiary structure of protease 3075 was predicted using PHyre2 and SWISS-MODEL programs. SWISS-MODEL software was used to create a 3D structure. To end this, the protease 3075 amino acid sequence, the obtained PDB file from SWISS-MODEL software, and the PDB file of several crystallographic proteins that revealed the most similarity and e-value with protease 3075, were introduced to EasyModeller software. After multiple alignments, the three-dimensional structure of the protein was simulated (Fig 5A). According to the obtained results, protease 3075 has a large number of salt bridges between $Gln_{59}$-$Lys_{38}$, $Asp_{156}$-$Arg_{152}$, $Gln_{40}$-$Lys_{37}$, $Gln_7$-$Arg_{80}$, $Asp_{139}$-$Asp_{152}$, $Asp_{145}$-$lys_{118}$

**Table 4. Biochemical properties of protease 3075.**

| Enzyme | Number of amino acids | Number of atoms | Molecular weight | pI | instability Coefficient | Coefficient of instability | Net charge |
|---|---|---|---|---|---|---|---|
| protease 3075 | 175 | 2711 | 19.12 | 6.73 | 39.89 | 89.20 | 0 |

Gln$_{13}$ Lys$_{45}$ residues (Fig 5B). Amino acid sequence analysis in the DISULFIND database did not show the disulfide band in protease 3075 amino acid sequences (S3 Fig). According to the obtained results from examining the amino acid sequence of the protease studied by PROSITE, it was found that this sequence overlaps with the C56 peptidase region of *PfpI* bacteria (S4 Fig). Moreover, the catalytic regions include histidine 104 and cysteine 103, where the

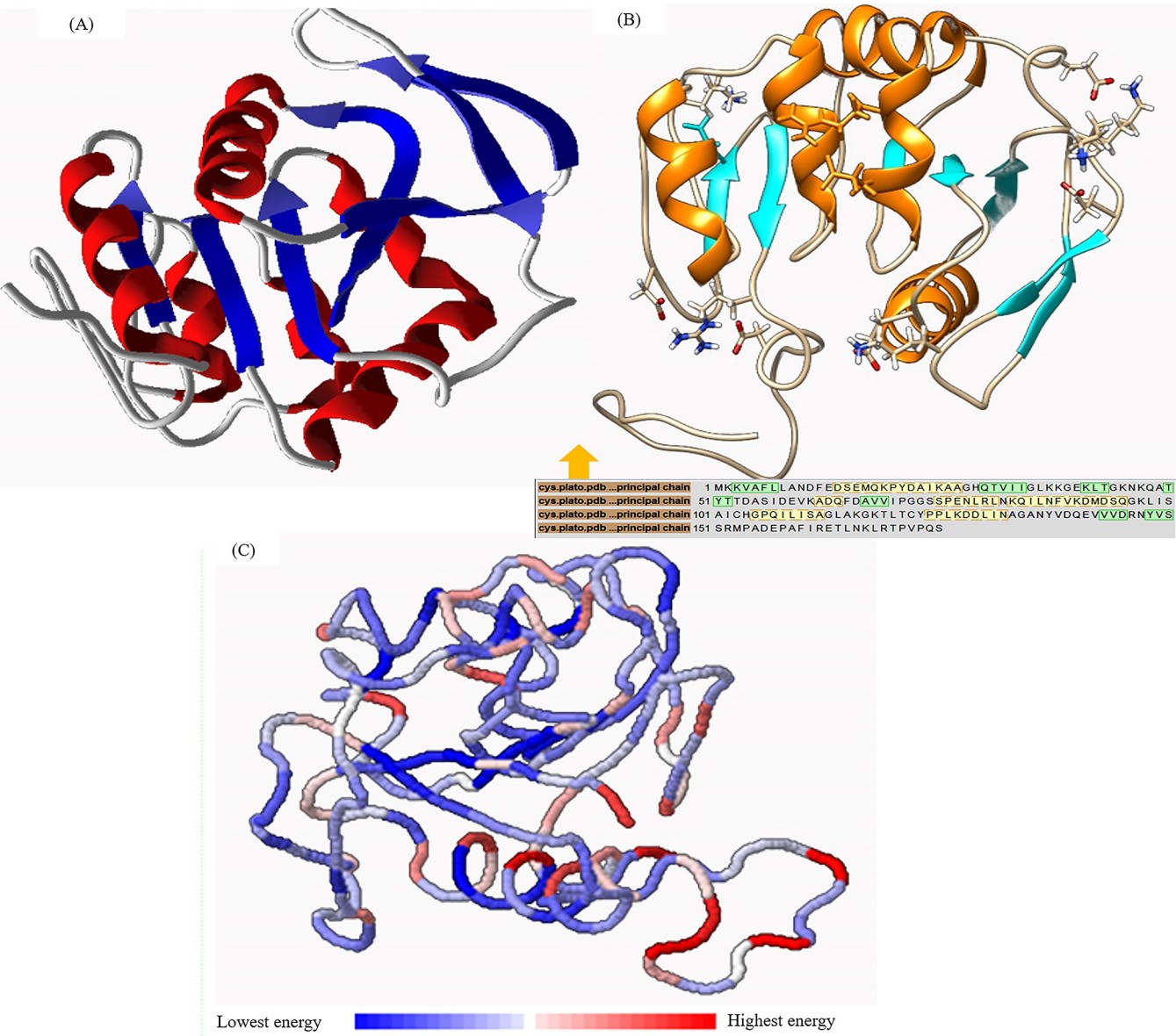

**Fig 5. The exhibition of protease 3075 structure.** (A): Salt bridges in the structure of protease 3075. (B):3D structure of the protein with 7 β-Sheets (blue) and 5 α-Helixes (red). (C): Prediction of the energy level of amino acids in protease 3075 structure.

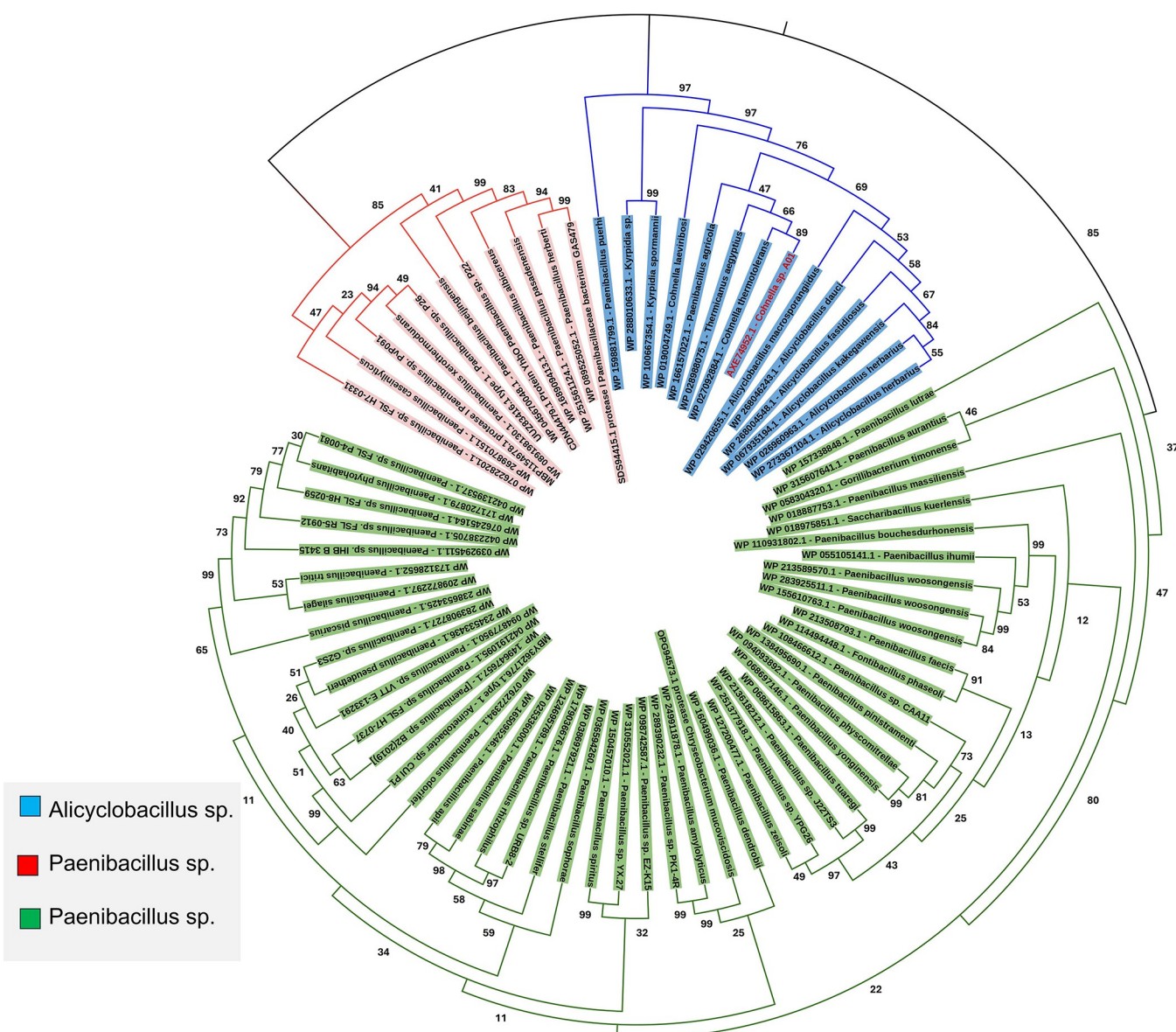

**Fig 6. Phylogenetic relationship of protease 3075 from *Cohnella* sp. A01 and its similar sequences.** The phylogenetic tree was constructed based on the alignment of the protein sequences with high similarities. The protease 3075 is marked with a red font.

nucleophilic attack was carried out by the cysteine in the amino acid sequence. The biochemical features of protease 3075 and many similar proteases were monitored by the ProtParam tool (Tables 1 and 2 in the S1 File). Moreover, the total energy level of amino acids was estimated. The total energy is shown in red and blue areas in Fig 5C. The energy of amino acids from blue to red indicates an increase in energy. As observed, the total energy level of amino acids was in the blue area, which revealed the overall stability of the structure. Fig 6 shows the amino acid sequence of protease 3075 from *Cohnella* sp. A01 (accession number: AXE74952.1) and the subsequent protein BLAST result were utilized to construct the phylogenetic tree. The neighbor-joining method was employed, with 1000 bootstrap replicates inferred for each region using MEGA11 software (displayed adjacent to the branches). GenBank accession numbers are provided alongside the species names. Phylogenetic tree and sequence

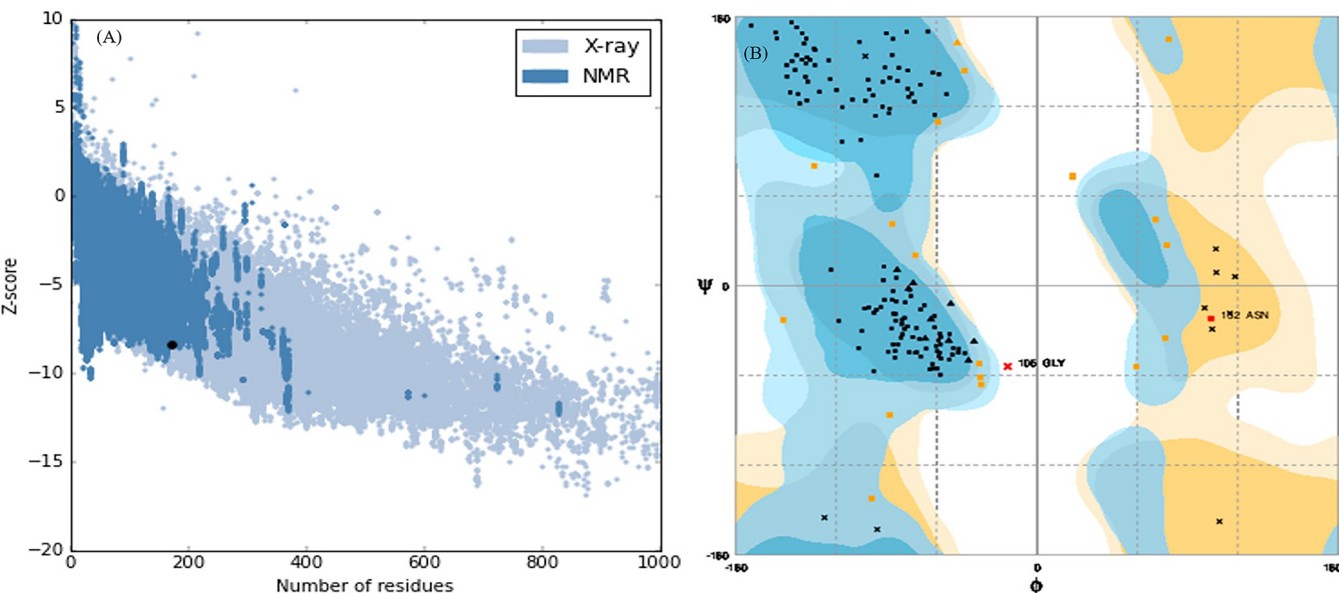

**Fig 7. The validation of modeled results.** (A): ProSA Z-score analysis for modeled protease 3075. The dark blue parts correspond to the NMR regions and the light blue parts correspond to the X-ray. (B): (B) Ramachandran's analysis. Blue areas indicate ideal amino acid placement, orange areas indicate acceptable areas and pale-yellow areas show unacceptable areas.

alignment analysis (Table 3 in the S2 File) discovered that protease 3075 had the most resemblance to the C56-*PfpI* superfamily (Fig 6).

**3.7.2. Crystallographic models, docking, and molecular dynamic simulation.** The ProSA-web result falls within the range of similar native protein scores, as indicated by a Z-score of -8.42 (Fig 7A). The Ramachandran plot confirmed the high quality of the final model. It showed that 98% of the amino acids in the protein structure were located in the allowed area, and the remaining 2% were in the acceptable range (Fig 7B).

Since Tween 80 had the greatest effect on the enzyme among the examined ligands, docking studies, and enzyme molecular simulation were performed in the presence of the mentioned ligand. Fig 8 shows the docking results of protease 3075 in the presence of tween 80. Fig 8A–8C represents the two-dimensional diagram of the Tween 80 interaction with protease 3075, and Fig 8D depicts the interaction mode between protease 3075 and Tween 80. Table 5 shows the hydrogen bond, free binding energy, final intermolecular energy, inhibition constant ($K_i$), and electrostatic energy values obtained from docking Tween 80 with protease 3075. The R-studio discovery program was used to assess the interaction between protease 3075 and Tween 80. The negative value of the free binding energy (-4.83 kCal.mol$^{-1}$) indicated that the enzyme and the ligand spontaneously interacted. The results of molecular docking showed that hydrogen bond interactions occurred between Tween 80 and $Lys_{97}$, $Lys_3$, $Gln_{64}$, $Asp_{66}$, and $Lus_{98}$. Vander Waals interactions were also observed between $Asp_{63}$, $Phe_{65}$, and $Val_{172}$. Fig 8D displays the hydrogen bond and van der Waals interactions. As observed, the hydrogen bond length was shorter than van der Waals. As a result, the ligand could be embedded in the secondary structure of protease 3075 by hydrogen bond and van der Waals interactions.

Molecular dynamics simulation methods provide detailed molecular information about the atom's trajectories to survey the structural stability, conformational changes, and flexibility of protease 3075 in accompaniment of Tween 80. To disclose the comprehensive effects of Tween 80 interaction with the protease and confirm the experimental data, MD simulation parameters including root mean square deviation (RMSD), gyration radius (RG), root mean square

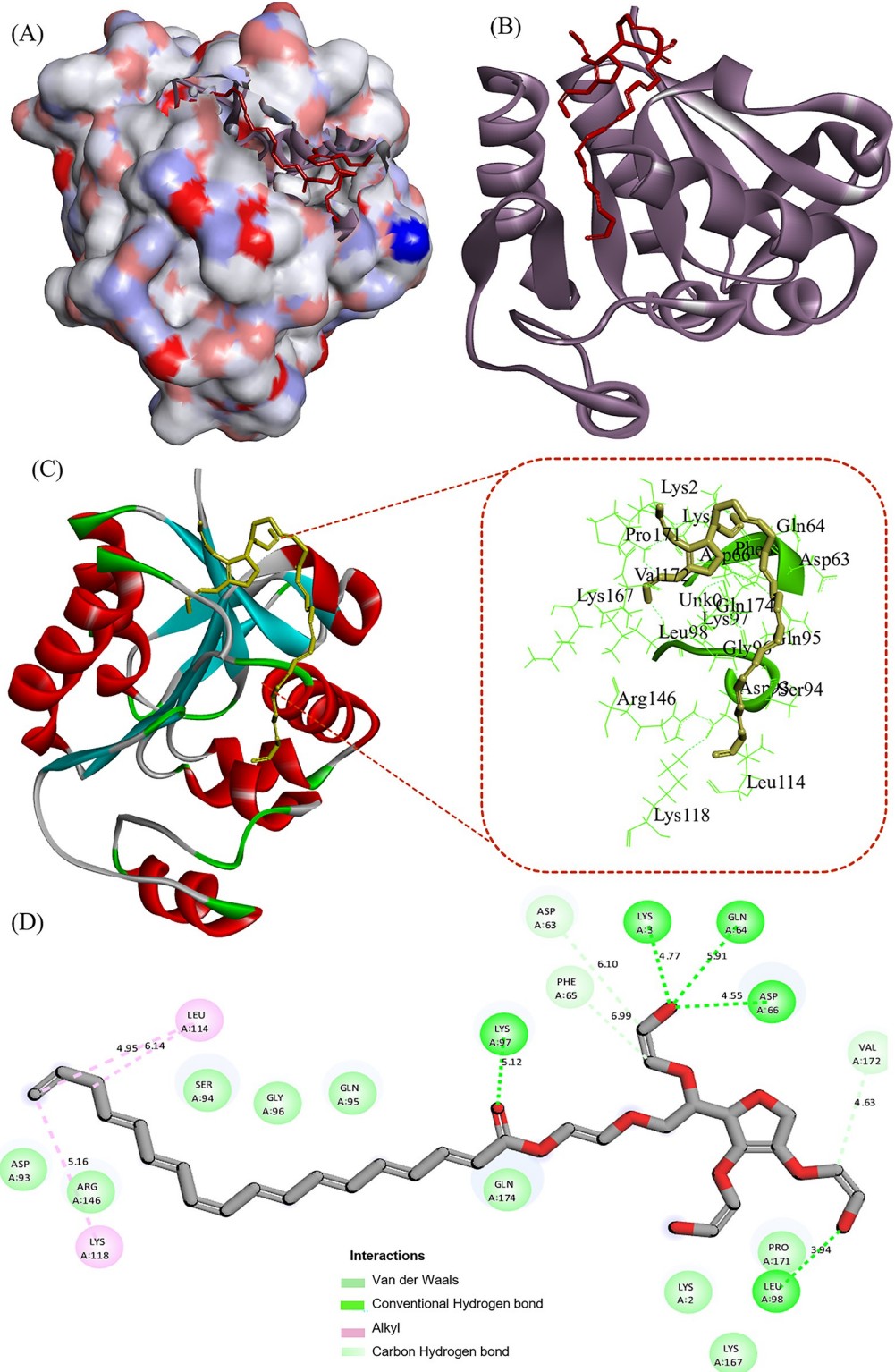

**Fig 8. Two-dimensional diagram of the Tween 80 interaction with protease 3075.** (A): The hydrophobic surface representation of Tween 80 and protease 3075 interaction. (B): Ribbon cartoon of Tween 80 and protease 3075 interaction. (C): Tween 80 binding site to protease 3075 amino acids residues. (D): Binding forces of tween 80 with protease 3075.

**Table 5. Results of the Tween 80 docking with protease 3075.**

| Free Binding energy (kcal/mol) | Inhibition Constant (298.15K), µM | Final Intermolecular energy (kcal/mol) | VDW + H-bond + dissolve energy (kcal/mol) | Electrostatic Energy (kcal/mol) | Final Total Internal Energy (kcal/mol) | Torsional Free Energy (kJ/mol) | Unbound System's Energy (kJ/mol) |
|---|---|---|---|---|---|---|---|
| -4.83 | 288.14 | -11.99 | -11.79 | -0.20 | -3.28 | 7.16 | -3.28 |

fluctuation (RMSF), and secondary structural changes were considered [33, 34]. The mentioned enzyme structural stability in the accompaniment of Tween 80 was studied using the root mean square deviation (RMSD) parameter. The average interval between the ligand and protease 3095 atoms in a started and balanced structure mode was examined over the RMSD analysis. According to the computed RMSD values, the developed model was appropriate for further analysis. The RMSD amounts of the free protease 3075 and Tween 80-protease 3075 complex during 100 ns have been listed in Table 6 and Fig 9A. protease 3075 structure and the highest score docking structure of Tween 80 with the enzyme were selected as reference structures during simulation analysis. The convergence of MD simulations on the mixtures of Tween 80 -protease 3075 was investigated after adaptation by the RMSD of atoms, which was specified by the Eq (5) [34, 35]:

$$\text{RMSD} = \sqrt{\frac{1}{N}\sum_{i=1}^{N}\left(r_i - r_{i,ref}\right)^2} \tag{5}$$

The RMSD calculated data revealed the sketched system by free protease and Tween 80-protease 3075 complex attained a steady-state mood. The stability information received from the enzyme showed that Tween 80 caused an increase in the protease stability in a complex form (0.281±0.047 nm) compared with the free enzyme (0.225±0.026) (Fig 9A).

The variations and the average distance between the atoms in a dynamic system due to the flexibility of the backbone of portions are measurable. Root Mean Square Fluctuation (RMSF) is one of the most popular methods of measuring protein flexibility and mobility [36]. The fluctuation of all residues for protease 3075 in water and the complexed form with Tween 80, during the period scale, were depicted in Fig 9B. The overall average RMSF amounts of each amino acid residue are listed in Table 6. The RMSF plot revealed a less structural fluctuation profile in protease 3075 complexed with Tween 80, compared to the protease 3075 in water.

When a protein suffers alterations in compaction, these variations can be evaluated by gyration radius (RG) [37]. The RG was employed to disclose the structural compactness and conformation of the protease over the MD simulation. The gyration radius of a protein is the root mean square distance of all atoms from the center of gravity, revealing the relation between compaction and folding of a protein. The assessments of fluctuations and alterations in the amount of protease compaction were evaluated by the gyration radius (RG), which was determined by the following equation [38]:

$$\text{RG} = \sum_{i=1}^{N} m_i(r_i - \bar{r})^2 \Big/ \sum_{i=1}^{N} m_i \tag{6}$$

**Table 6. The average and standard deviations of RMSD, RMSF, and RG.**

| Protein | RMSD (nm) | RG (nm) | RMSF (nm) | H-bond pro-sol (nm) | H-bond pro-pro (nm) |
|---|---|---|---|---|---|
| Protease -water | 0.225±0.026 | 1.491±0.012 | 0.121±0.086 | 28.15±3.49 | 6.60±1.67 |
| Protease -tween 80 complex | 0.281±0.047 | 1.478±0.007 | 0.114±0.070 | 29.34±3.50 | 5.60±1.67 |

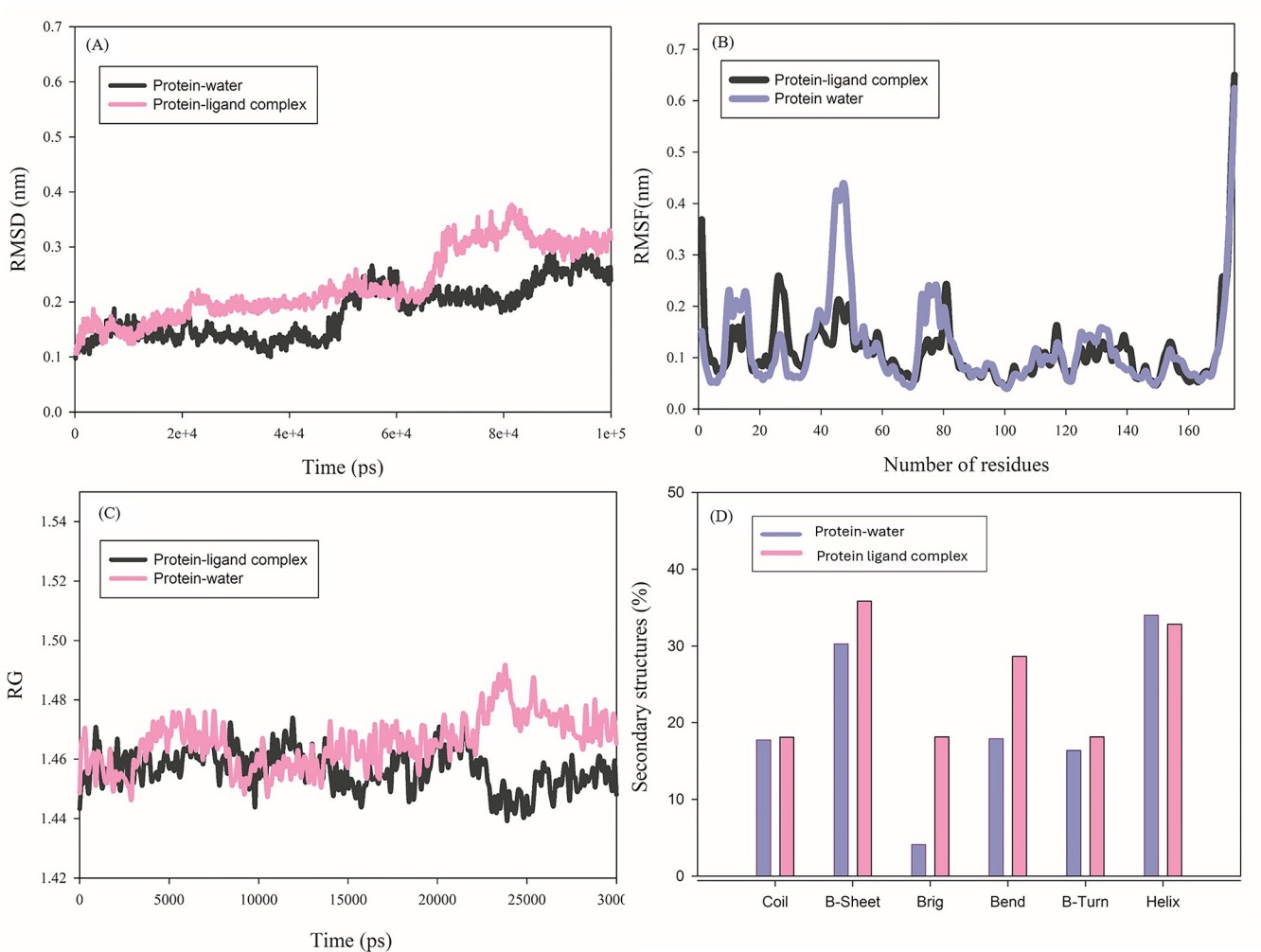

**Fig 9. The MD simulation parameters of free protease 3075 (protein-water) and complex form with Tween 80.** (A): RMSD plot, (B): RMSF plot, (C): RG plot in the water micro-environment and complex with Tween 80, and (D): the secondary structure content of protease 3075 in the presence and absence of Tween 80.

The RG consequence of the modeled system by protease 3075 and Tween 80 has been shown in Fig 9C. As observed the RG values of the protease 3075-Tween 80 complex reached a higher level compared to the native enzyme. Thus, accompaniment with Tween 80 caused a reduction in the protease compaction. The index values of RG are presented in Table 6.

Over the MD simulation, the secondary structure content of protease 3075 to disclose the enzyme structure was done. By calculating the secondary structure of the mentioned protease in the presence and absence of Tween 80, the conformational alterations were examined in more detail. The enzyme structure was identified by the presence of 32.84% α-Helix, 35.85% β-sheet, 18.16% β-turn, and 18.11% random Coil (Table 7). The dominant secondary structures of protease 3075 were β-sheet and α-Helix. The α-Helix content of the enzyme increased by 34.00% as a result of protease 3075 interaction with Tween 80. In contrast, the content of β-turn and β-sheet decreased by 16.37% and 30.28%, respectively (Table 7 and Fig 9D). The results of RMSF and RG suggested the protease after interaction with Tween 80 reached a more flexible conformation. The secondary structure analysis was in agreement with RMSF

**Table 7. Secondary structure content of the native protease 3075 and complex form with Tween 80.**

| protein | α-Helix (%) | β-sheet (%) | β-Bridge (%) | β-Turn (%) | Random Coil (%) | Bend (%) |
|---|---|---|---|---|---|---|
| Protease-water | 32.84 | 35.85 | 18.16 | 18.16 | 18.11 | 28.66 |
| Protease -Tween 80 complex | 34.00 | 30.28 | 4.11 | 16.37 | 17.76 | 17.92 |

and RG results. Furthermore, the formation of hydrogen bonds in protein-protein and protein-solvent states was investigated before and after ligand binding. The results showed that after the binding of the ligand, the amount of hydrogen bonds decreased in the protein-protein (pr-pr) states compared to the protein-solvent (pr-sol) state. This finding showed an increase in the level of flexibility of protease 3075 in the presence of Tween 80, which was in agreement with the results of RMSF, RG, and secondary structure analysis.

## 4. Discussion

Nowadays due to significant industrial requirements, researchers have been interested in screening new proteases with improved properties. Because of some beneficial features including rapid growth, small space for cultivation, and large diversity, micro-organisms (bacteria and fungi) are important sources of naturally occurring proteases. Bacterial proteases have many advantages compared to Fungai including better thermal stability and reaction rates [39, 40]. So far various protease-encoding bacterial genes with the purpose of engineering, characterization, and overproduction have been expressed in new hosts [41–44]. In the current study, we isolated a novel enzyme called protease 3075 from a thermophilic *Cohnella* sp A01. Then recombinantly expressed, purified, and discovered the biochemical and structural characterization of the protease by in-silico and in-vitro analysis. First in silico analysis was performed. According to the computer results, the translated sequence of the enzyme revealed that protease 3075 was produced intracellularly with a high aliphatic index. This is a characteristic feature of thermophilic bacterial enzymes. Moreover, the obtained results from the analysis of the amino acid sequence of the enzyme by the BLASTP analysis showed the protease sequence overlaps with the C56 peptidase region (ranging from amino acid 2 to 169) of *PfpI* bacteria. This protease belongs to a large family that almost has a member in all organisms. Despite the extensive enzymatic activities, this superfamily plays various functions such as neuron protection and chaperons. However numerous members of this family have been classified, but their bioinformatic and biochemical properties have not been fully studied. The enzymatic activity of protease 3075 was done by the thiol side chain of cysteine. Homology modeling was done by Modeller 9V7, and then the quality of the obtained 3D structure was confirmed by the Ramachandran plot. 98% of the amino acid residues of the protease structure were in the allowed regions and the remaining 2% were in the acceptable range. ProSA was used to determine the Z-Score point and protein energy balance. The computed Z-score (-8.42) indicated the protein structure is in the overlapping region of NMR and X-ray. Thus, based on the obtained these results the predicted structure was highly accurate and reliable. The evaluation of enzyme activity in the presence of different additives showed that the activity of protease 3075 increased significantly in the presence of tween 80 and acetone. Tween 80 had significant effects on protease 3075 activity compared to other additives. Since, the mechanism action of the surfactant on the enzyme structure and function has not been completely investigated, therefore, to confirm the experimental results the docking and molecular dynamic simulation were done. The docking results revealed that Tween 80 interacted with the active site of protease 3075 by Hydrogen bond and van der Waals forces in the reaction mixture. The RMSD amounts of protease 3075-Tween 80 complex was 0.281±0.047, revealing more stable

conformation of the enzyme after interaction with Tween 80. Simultaneously, the RMSF and RG values showed an increase in the flexibility of the enzyme in the presence of Tween 80. The secondary structure studies showed that Tween 80 decreased the content of the β-sheet and increased the α-helix amounts. This finding was consistent with the values obtained from RMSF and RG. The interaction between protease 3075 and Tween 80 stayed approximately stable after 100 ns of simulation and strengthened with several new contacts between the oxygen group of the ligand and $Lys_{97}$, $Lys_3$, $Gln_{64}$, $Asp_{66}$, and $Lus_{98}$, $Asp_{63}$, $Phe_{65}$, and $Val_{172}$ of the enzyme. Moreover, after the interaction of Tween 80 with the enzyme's amino acids, the amounts of protein-protein and protein-solvent hydrogen bonds, and bond length were changed. These results explain why tween 80 was affected by the activity of protease 3075. By this mechanism, the mentioned surfactant affects the physicochemical properties of protease 3075.

The modeled structure of protease 3075 revealed the enzyme made of α and β sandwich tertiary structure, which is existent in the recognized structure of *PfpI* superfamily representative. Moreover, the catalytic regions of the enzyme include $His_{104}$ and $cys_{103}$, which were subjected to nucleophilic attack by the cysteine amino acid. The predicted tertiary structure by PHyre2 and SWISS-MODEL programs showed that protease 3075 contained 5 alpha helixes and 7 beta sheets, which were in strong agreement with the secondary structures predicted by the Psipred website (S1 Fig).

The molecular weight of most bacterial proteases is between 15–45 kDa [45]. As observed in Fig 6, the recombinant protease 3075 was identified at nearly 19 kDa position, that confirmed by Western blot and zymography analysis. The His-tag affinity column chromatography was used to purify the produced protease. SDS-PAGE analysis showed that the purified enzyme was homogenous. After purification of the recombinant enzyme, structural and biochemical characteristics, the effect of additives, different temperatures, and pH on enzyme activity were investigated. Since protease 3075 showed the highest specificity for casein, Therefore, casein was used as the enzyme substrate. Protease 3075 revealed approximately more than 40% relative activity at pH 5–8. Moreover, the enzyme activity remained more than 70% in the temperature range of 10–70 degrees. Protease 3075 revealed supreme tolerance to a wide range of pH and high temperatures. The highest activity of protease 3075 was at 70˚C and a pH of 6. The enzyme stability and activity during a wide pH range and temperatures potentially make protease 3075 useful for various industrial applications. Therefore, investigation of the thermodynamic parameters of the protein supplied insights into the factors affecting protease 3075 stability. As shown, protease 3075 has intrinsic structural stability that its activity maintained at high temperatures. The activation energy ($E_a$) is the minimum value of energy that is required to activate atoms or molecules for the reaction. Reduction in the amounts of $E_a$ causes increases in the fraction of reactant molecules that provide sufficient energy to product formation. The maximum amounts of entropy and enthalpy express the efficiency of the transitional state and the negative values of free energy reveal the spontaneity of a reaction [46]. In the current study, we analyzed the protease 3075 thermodynamic parameters including $Ea^{\ddagger}$, $\Delta G^{\ddagger}$, $\Delta H^{\ddagger}$, $\Delta S^{\ddagger}$, and $\Delta G^{\ddagger}$. One of the key factors in understanding the enzyme's thermal stability is the activation energy. The activation energy causes the thermal inactivation of an enzyme. High values of $Ea^{\#}$ require more energy to denature the enzyme, which suggests higher thermal stability [47, 48]. The low amounts of $Ea^{\ddagger}$, $\Delta H^{\ddagger}$, and $\Delta S^{\ddagger}$ at the optimum temperature revealed the efficient transition state of $ES^{\ddagger}$ at 70˚C. High $\Delta G^{\#}$ values at 70˚C indicated that protease 3075 was resistant to transition state unfolding and required high inactivation energy for denaturation. Moreover, the transition state of protease 3075 was higher at the optimal temperature. Bioinformatics analysis showed that protease 3075 has the most similarity with the $C_{56}$ peptidase region of *PfpI* bacteria. The activity pattern of protease 3075 at different pH and

temperatures showed that the enzyme was most active at approximately 70 ˚C and pH 6. To compare the enzyme's different substrates, optimum temperature, and pH with other proteases, a summary of identified bacterial proteases from various sources was supplied in S1 and S2 Tables. Several documents have reported some ions such as metals play stabilizing impacts on enzyme activity and structure [49–51]. Metal ions have different effects on protease activity [52, 53]. A limitation regarding cysteine proteases is that against metal ions their activities are rapidly suppressed [54–56]. Hence, such proteases need chelating agents and mild reductants so they are not economical [57]. In contrast, serine proteases that do not have such restrictions can be regarded as appropriate candidates for industrial applications. In the current study, the impacts of 8 metal ions were investigated at optimum temperature and pH. The obtained results revealed that despite some cysteine proteases, protease 3075 is approximately stable in various concentrations of metal ions. Among metal ions, CaCl2 at a concentration of 2 mM increased the enzyme activity 2-fold. Enzyme activity in the presence of organic solvents at different concentrations and under optimal conditions showed that acetone 2% increased enzyme activity by about 150 times. Methanol had the most negative effect on the enzyme activity so the relative activity of the enzyme reached less than 50%. In contrast, isopropanol 2% had almost no effect on enzyme activity. Chemical compounds such as iodoacetic acid and iodoacetamide as inhibitors of cysteine protease activity strongly suppressed the activity of protease 3075, while PMSF and EDTA decreased the enzyme activity by about 50%. Tween80, as a surfactant, increased the protease activity 4-fold, while other surfactants inhibited the enzyme activity. Finally, regarding the remarkable stability and biochemical properties of the novel extracted enzyme, protease 3075 can be introduced for various industries, including detergent industries.

## Supporting information

**S1 Fig.** M: molecular size marker, 1: genomic DNA extracted from *Cohnella* sp. A01, 2: circular plasmid, 3: Digested recombinant plasmid, 4: PCR product of the desired gene, 5: enzymatic digestion of recombinant plasmid, 6: empty digested vector with Nde I enzyme, 7: PCR colony, 8: PCR plasmid.
(PPTX)

**S2 Fig. Secondary structure was drawn by Psipred.** Yellow arrows represent beta sheets, and pink cylinders represent alpha helix, which are connected by coils.
(PPTX)

**S3 Fig. Monitoring the number of disulfide bonds (DB).** Registering the number one and zero against DB_state indicates the presence and absence of disulfide bonds, respectively. DB_conf represents the confidence factor. The closer the DB_conf number to 9, the higher the confidence level.
(PPTX)

**S4 Fig. The results of the protein sequence check on the PROSITE website.**
(PPTX)

**S1 Table. Comparison of protease 3075 with other industrial proteases in terms of effects on different substrates.**
(DOCX)

**S2 Table. Comparison of some biochemical features of protease 3075 with other proteases.**
(DOCX)

**S3 Table. Sequence similarity analysis of protease 3075.**
(DOCX)

## Acknowledgments

The authors are grateful to the National Institute of Genetic Engineering and Biotechnology (NIGEB), Iran for providing research facilities.

## Author Contributions

**Conceptualization:** Saeed Aminzadeh.

**Data curation:** Fatemeh Hashemi Shahraki, Narges Evazzadeh.

**Formal analysis:** Fatemeh Hashemi Shahraki, Saeed Aminzadeh.

**Funding acquisition:** Narges Evazzadeh.

**Investigation:** Narges Evazzadeh.

**Methodology:** Fatemeh Hashemi Shahraki, Narges Evazzadeh.

**Project administration:** Saeed Aminzadeh.

**Software:** Narges Evazzadeh.

**Supervision:** Saeed Aminzadeh.

**Validation:** Saeed Aminzadeh.

**Writing – original draft:** Fatemeh Hashemi Shahraki.

**Writing – review & editing:** Fatemeh Hashemi Shahraki, Saeed Aminzadeh.

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
