## [Decision Letter · Decision Letter 0]

10 Jul 2024

PONE-D-24-19039Heterologous expression, purification, and biochemical characterization of protease 3075 from Cohnella sp. A01PLOS ONE

Dear Dr. Aminzadeh,

Thank you for submitting your manuscript to PLOS ONE. After careful consideration, we feel that it has merit but does not fully meet PLOS ONE’s publication criteria as it currently stands. Therefore, we invite you to submit a revised version of the manuscript that addresses the points raised during the review process. Both reviewers identified several major concerns with the manuscript in its current form. In particular, please be certain to clearly describe how this study differentiates itself from prior work. PLOS One publication criteria states that "If a submitted study is very similar to previous work, authors must provide a sound scientific rationale for the submitted work". As stated by reviewer 2, the observation that Tween 80 stabilizes this protein is relatively derivative of many prior works. If you wish to include this data, it is important that you justify these results as distinct from prior observations, being sure to reference and discuss the existing literature. Please also ensure that you consider and include comparisons with previously characterized proteases and, as indicated above, include the appropriate literature references. In addition, both reviewers strongly recommended thorough editing for language to ensure that the information is presented clearly. The language must be clear, correct, and unambiguous to meet PLOS One standards.

We look forward to receiving your revised manuscript.

Kind regards,

Jarrod B. French, PhD

Academic Editor

PLOS ONE

Reviewers' comments:

Reviewer's Responses to Questions

**Comments to the Author**

1. Is the manuscript technically sound, and do the data support the conclusions?

Reviewer #1: Yes

Reviewer #2: Yes

2. Has the statistical analysis been performed appropriately and rigorously? 

Reviewer #1: N/A

Reviewer #2: Yes

3. Have the authors made all data underlying the findings in their manuscript fully available?

Reviewer #1: Yes

Reviewer #2: Yes

4. Is the manuscript presented in an intelligible fashion and written in standard English?

Reviewer #1: No

Reviewer #2: Yes

5. Review Comments to the Author

Reviewer #1: Heterologous expression, purification, and biochemical characterization of protease

from Cohnella sp. A01

In this manuscript, the authors described the cloning and expression of a protease from a

thermophilic organism Cohnella sp. A01. In this study, the biochemical and structural aspects of protease were investigated, and it was described as belonging to the C56 - PfpI

superfamily. The gene was cloned and expressed in E. coli BL21. The activity was increased by 4-fold in Tween 80 and acetone, a fact stated to be mediated by the hydrogen bonds and Van der Waals forces. Based on the Molecular dynamics simulations (MDS), Tween 80 heightened the stability of the structure.

Specific Comments

1. The initial part of the abstract is long and unnecessary. The background can be condensed into one set of sentences.

2. The biochemical and structural aspects of the protease were investigated for which enzyme, the native or recombinant proteases?

3. The results of the expression should be briefly mentioned in the abstract.

4. Tween 80 and acetone increased the enzyme activity by fourfold. However, the sentence in the abstract is unclear and ambiguous.

5. The sentence in the abstract-----"presence of Tween 80 heightened the stability of the protease structure,” is ambiguous and appears speculative. It needs further clarity. What is the indication of “heightened stability” in the context of the molecule's structure?

6. The abstract overall needs reorganization and condensation.

7. In the methods, why the section, ‘2.2.5. Cohnella sp. A01 cell culture and DNA extraction’ is included at a later stage? In this context, the authors should reconsider the organization and sequence of the contents. Should it be started with bioinformatics and its analysis?

8. What is the nature of the protease gene primer? How was it obtained?

9. In the table, the number of atoms for the entire protease molecule? What is its significance in the context of structure and function?

10. To conclude, the structural stability of the protease in response to its interaction with twin-80 needs to be described in a condensed and cohesive manner.

11. If certain studies with the native enzymes are also done, a comparison of the enzymatic/biochemical feature should also be discussed.

12. Some figures are placed at the end, while the related captions are included in the text. It would be better and more convenient if the captions were placed near/below the figures—for instance, Figure S4.

13. Marker bands are fuzzy. Well-3 and 5 are both labeled as digested recombinant plasmid, but their band patterns are different. This requires confirmation and clarity.

14. What are the bands inside the well-7 and 8? Are they supposed to be visualized, or is it a false result?

15. Why is the band in well-8, PCR plasmid at lower bp to colony PCR band? Shouldn’t it be at a higher bp?

16. The arrangement of figures and their captions are scattered.

17. Some specific distinctions regarding structure-function, gene cloning and expression, and nobility between this enzyme and others described so far should be highlighted in the highlights and the conclusion/abstract. Otherwise, the study would appear as routine. Some more recent work on these aspects should also be included, specifically from extremophiles.

18. The manuscript over all needs careful proofreading for typographical errors and improvement in language and expression.

Reviewer #2: Proteases are interesting and important enzymes and it is well worth to study. In this ms, the authors was cloned and heterologously expressed the protease 3075 gene in Escherichia coli and some properties of enzymes were studied. The innovation of the article is still insufficient. And I have several comments on the ms.

1. The ms should clearly be edited by an English spoken person.

2. The authors should indicate the source of Cohnella sp.A01 in the manuscript.

3. Tween 80 is a commonly used surfactant and it can stabilize proteins and improve the catalytic activity of enzymes. What is the significance of the molecular docking between Tween 80 and protease AND MD?

4. The unit format in the article should be uniform. e.g. Line 162 “ng/μl” , Line 212 “ mg.ml-1”)

5. Line 212, 213 “ mg.ml-1” should be “ mg·ml-1”

6. K and V in Km, Vmax, Kcat should be italicized

7. Line 230 “ The reaction mixture was incubated for 3 hours at 4 ºC.” Why did the author choose 4 ºC?

8. Section 3.1.1. Should provide sequence similarity analysis of protease 3075.

9. Fig or Fig. ? It should be unified

10. Section 3.4. The specific activity of protease 3075 should be provided.

11. Table 7, Why some substances that strongly inhibit enzyme activity at low concentrations are less so at high concentrations?

12. None of the pictures are clear enough.

6. PLOS authors have the option to publish the peer review history of their article (what does this mean?). If published, this will include your full peer review and any attached files.

Reviewer #1: **Yes: **Satya P. Singh

Reviewer #2: No

---

## [Author Response · Author response to Decision Letter 0]

19 Aug 2024

Please submit your revised manuscript by Aug 24 2024 11:59PM

Reviewer #1

1. The initial part of the abstract is long and unnecessary. The background can be condensed into one set of sentences

Thank you for pointing this out! As mentioned, the background was condensed into one set of sentences.

2. The biochemical and structural aspects of the protease were investigated for which enzyme, the native or recombinant proteases?

Thank you! The biochemical and structural aspects of the protease were investigated for recombinant proteases 3075.

3. The results of the expression should be briefly mentioned in the abstract.

Thank you for pointing this out! We rewrite the abstract part.

4. Tween 80 and acetone increased the enzyme activity by fourfold. However, the sentence in the abstract is unclear and ambiguous.

Thank you! We found and corrected it.

5. A-The sentence in the abstract-----"presence of Tween 80 heightened the stability of the protease structure,” is ambiguous and appears speculative. It needs further clarity. 

 B- What is the indication of “heightened stability” in the context of the molecule's structure?

A- Thank you! We improved the abstract part. 

B- There was a typographical error. We considered and corrected it.

6. The abstract overall needs reorganization and condensation.

Thank you for pointing this out! As mentioned, the abstract part was reorganized.

7. In the methods, why the section, ‘2.2.5. Cohnella sp. A01 cell culture and DNA extraction’ is included at a later stage? In this context, the authors should reconsider the organization and sequence of the contents. Should it be started with bioinformatics and its analysis?

Thank you for pointing this out! As mentioned, we reconsider the organization and sequence of the contents.

8. What is the nature of the protease gene primer? How was it obtained?

Thank you! The nature of the protease gene primer was a deoxyribonucleic acid sequence. The primers were made by the Tekapozist company in Iran.

9. In the table, the number of atoms for the entire protease molecule? What is its significance in the context of structure and function?

Thank you! Generally, the number of atoms in the protein was determined to assign the size and complexity of the protein. Moreover, to perform molecular docking and MD simulation studies it was better to know the number of atoms to design the simulation box.

10. To conclude, the structural stability of the protease in response to its interaction with twin-80 needs to be described in a condensed and cohesive manner.

Thank you! As advised, we described the structural stability of protease 3075 in the presence of Tween-80 in a condensed and cohesive manner (472-475 lines).

11. If certain studies with the native enzymes are also done, a comparison of the enzymatic/biochemical feature should also be discussed.

Thank you for pointing this out! As mentioned in the manuscript (line 394), we comprised the temperature, optimum pH, substrate specificity, and some features of protease 3075 with other proteases. Data added to the supplementary file (S5 and S6 Tables).

12. Some figures are placed at the end, while the related captions are included in the text. It would be better and more convenient if the captions were placed near/below the figures—for instance, Figure S4.

Thank you for pointing this out! Because S4 Fig was related to supplementary files, it was placed in the supplementary file. But as mentioned, we rearranged the other figures According to the mentioned order.

13. Marker bands are fuzzy. Well-3 and 5 are both labeled as digested recombinant plasmid, but their band patterns are different. This requires confirmation and clarity.

Thank you! At that time, the quality of the used camera was not suitable. According to our knowledge, their band patterns are not different. Unfortunately, the gel is now gone and it is not possible to take pictures again. However, we had another photo of the same gel and replaced it.

14. What are the bands inside the well-7 and 8? Are they supposed to be visualized, or is it a false result?

Thank you! There is only one band in each mentioned well. As described in the figure caption, the bands inside wells 7 and 8 represent the PCR colony and PCR plasmid, respectively.

15. Why is the band in well-8, PCR plasmid at lower bp to colony PCR band? Shouldn’t it be at a higher bp?

Thank you! The bands in Wells 7 and 8 are in the same area (525 bp), which is indicated by the red line on the gel image (Fig. 1). However, because, in Well 7 a little bit of the bacterial colony was visually picked up, compared to well 8 (PCR plasmid), its band is only slightly sharper.

Fig. 1: Gel electrophoresis.7:PCR colony, 8: PCR plasmid.

16. The arrangement of figures and their captions are scattered.

Thank you for pointing this out! As mentioned in the answer to question 12, We rearranged the figures.

17. Some specific distinctions regarding structure-function, gene cloning and expression, and nobility between this enzyme and others described so far should be highlighted in the highlights and the conclusion/abstract. Otherwise, the study would appear as routine. Some more recent work on these aspects should also be included, specifically from extremophiles.

Thank you for pointing this out! We improved the abstract and conclusion parts.

18. The manuscript over all needs careful proofreading for typographical errors and improvement in language and expression.

Thank you! We found typographical errors and improved the language of the manuscript.

Reviewer #2:

1. The ms should clearly be edited by an English spoken person.

Thank you! We improved the language of the manuscript.

2. The authors should indicate the source of Cohnella sp.A01 in the manuscript.

Thank you! The bacterial genome of Cohnella sp. A01 was isolated from a shrimp breeding pond in Abadan, Iran. This sequence was determined by Baseclear Netherlands company and its possible genes were predicted

3. Tween 80 is a commonly used surfactant and it can stabilize proteins and improve the catalytic activity of enzymes. What is the significance of the molecular docking between Tween 80 and protease AND MD?

Thank you ! As mentioned in the manuscript (423-425 lines), since Tween 80 had the had best binding energy and the greatest effect on the enzyme among the examined ligands, docking studies, and enzyme molecular simulation were performed in the presence of the mentioned ligand.

4. The unit format in the article should be uniform. e.g. Line 162 “ng/μl” , Line 212 “ mg.ml-1”)

Thank you for pointing this out! All unit formats were uniform.

5. Line 212, 213 “ mg.ml-1” should be “ mg·ml-1”

Thank you! We found and corrected it.

6. K and V in Km, Vmax, Kcat should be italicized

Thank you for pointing this out! We italicized Km, Vmax, and Kcat.

7. Line 230 “ The reaction mixture was incubated for 3 hours at 4 ºC.” Why did the author choose 4 ºC?

Thank you! In general, after the preparation of the reaction mixture, to prevent the loss of activity or any type of additional reaction, the mixture was kept at 4 ºC. 

8. Section 3.1.1. Should provide sequence similarity analysis of protease 3075.

Thank you for pointing this out! Sequence similarity analysis of protease 3075 prepared in Excel format in a supplementary file.

9. Fig or Fig. ? It should be unified

Thank you for pointing this out! We found and unified its writing format.

10. Section 3.4. The specific activity of protease 3075 should be provided. Specific enzyme activity is the number of enzyme units per ml divided by the concentration of protein in mg/ml. 

Thank you! Specific activity values for crud extract and after Ni-NTA are reported in Table 1.

Total Protein (mg.ml-1) Total Activity (U.ml-1) Specific Activity (U.mg-1 protein)

11.2 3.92 3.92/11.2= 0.35

3.6 1.44 1.44/3.6= 0.4

11. Table 7, Why some substances that strongly inhibit enzyme activity at low concentrations are less so at high concentrations?

Thank you for pointing this out! You are right. There was a typographical error. We considered and corrected it.

12. None of the pictures are clear enough.

Thank you! All experimental figures were plotted by sigma plot software. Figures were saved in TIFF format and 600 dpi. Moreover, bioinformatic figures were exported from the software.

---

## [Decision Letter · Decision Letter 1]

9 Sep 2024

Heterologous expression, purification, and biochemical characterization of protease 3075 from Cohnella sp. A01

PONE-D-24-19039R1

Dear Dr. Aminzadeh,

We’re pleased to inform you that your manuscript has been judged scientifically suitable for publication and will be formally accepted for publication once it meets all outstanding technical requirements.

Kind regards,

Jarrod B. French, PhD

Academic Editor

PLOS ONE

Reviewers' comments:

Reviewer's Responses to Questions

**Comments to the Author**

1. If the authors have adequately addressed your comments raised in a previous round of review and you feel that this manuscript is now acceptable for publication, you may indicate that here to bypass the “Comments to the Author” section, enter your conflict of interest statement in the “Confidential to Editor” section, and submit your "Accept" recommendation.

Reviewer #1: All comments have been addressed

2. Is the manuscript technically sound, and do the data support the conclusions?

Reviewer #1: Yes

3. Has the statistical analysis been performed appropriately and rigorously?

Reviewer #1: N/A

4. Have the authors made all data underlying the findings in their manuscript fully available?

Reviewer #1: Yes

5. Is the manuscript presented in an intelligible fashion and written in standard English?

Reviewer #1: Yes

6. Review Comments to the Author

Reviewer #1: The authors' responses and the revised manuscript appear satisfactory. I also noticed that another reviewer's comments were addressed.

7. PLOS authors have the option to publish the peer review history of their article (what does this mean?). If published, this will include your full peer review and any attached files.

Reviewer #1: **Yes: **Satya P. Singh

---

## [Editor Report · Acceptance letter]

16 Sep 2024

PONE-D-24-19039R1 

PLOS ONE

Dear Dr. Aminzadeh, 

I'm pleased to inform you that your manuscript has been deemed suitable for publication in PLOS ONE. Congratulations! Your manuscript is now being handed over to our production team.

Kind regards, 

on behalf of

Professor Jarrod B. French 

Academic Editor

PLOS ONE